# Does Pitavastatin Therapy for Patients with Type 2 Diabetes and Dyslipidemia Affect Serum Adiponectin Levels and Insulin Sensitivity?

**DOI:** 10.3390/jcm11226756

**Published:** 2022-11-15

**Authors:** Jeongmin Lee, Min-Hee Kim, Jung-Min Lee, Sang-Ah Chang

**Affiliations:** Division of Endocrinology and Metabolism, Department of Internal Medicine, Eunpyeong St. Mary’s Hospital, College of Medicine, The Catholic University of Korea, Seoul 03312, Republic of Korea

**Keywords:** dyslipidemia, diabetes mellitus, hydroxymethylglutaryl-CoA reductase inhibitors, adiponectin, insulin resistance

## Abstract

(1) Background: We aimed to demonstrate the effects of pitvastatin therapy on the serum levels of total adiponectin and high-molecular-weight (HMW) adiponectin in type 2 diabetes and the correlation with insulin sensitivity. (2) Methods: This study was designed as an open-labelled randomized trial. Patients with diabetes who were prescribed pitavastatin therapy were enrolled and randomized to either treatment with 2 mg of pitavastatin once daily (*n* = 44) (PITA group) or diet and exercise only, except their antidiabetic medications (*n* = 49), for 24 weeks. (3) Results: In lipid profiles, the reduction in total cholesterol (TC) and low-density lipoprotein cholesterol (LDL-C) was significantly increased in the PITA group (TC; 207.5 ± 20 vs. 195.5 ± 30.0 ng/dL, *p* < 0.001, LDL-C; 132.0 ± 15.8 vs. 123.1 ± 25.7 mg/dL, *p* < 0.001). Adiponectin and HMW adiponectin were elevated in the PITA group, compared to the control group without significance. The PITA group showed a lower level of HOMA-IR and HOMA-β levels. However, there was no significance (HOMA-IR; *p* = 0.5921 -at 12 weeks and *p* = 0.3645 at 24 weeks; HOMA-β; *p* = 0.8915 at 12 weeks and *p* = 0.7313 in 6 months). (4) Conclusions: The present study did not show a significant change in serum adiponectin or HMW adiponectin from baseline in serum adiponectin following pitavastatin therapy. Although statin has been considered as a risk for dysglycemia, pitavastatin did not affect insulin sensitivity.

## 1. Introduction

Adiponectin is the most abundant serum protein of 30 kDa derived from white adipose tissue [1]. Adiponectin plays a crucial role in atherogenesis and inflammation [2]. Lower serum adiponectin levels are correlated with insulin resistance pathophysiology and the development of type 2 diabetes mellitus (DM) due to an overexpression of inflammatory cytokines [3,4], and cardiovascular disease after vascular damage from the loss of proliferative smooth muscle cells and anti-atherogenic properties [5]. Adiponectin is synthesized as a monomer and assembles in homo-oligomers in low-molecular-weight trimeric, medium-molecular-weight hexameric, and high-molecular-weight (HMW) forms [6]. Adiponectin undergoes a complex post-translation process that is important for the formation and secretion of adiponectin multimers [7]. Different adiponectin is important for the formation and secretion of adiponectin multimers [7]. Different adiponectin complexes do not convert to other classes after secretion. Therefore, different molecular weights and their assembly are associated with the biological function of adipokines [8]. Among these adiponectins, HMW adiponectin plays a major role in insulin sensitivity and is a risk factor for obesity-related disease [9].

Regarding DM management, maintaining an ideal body weight and lowering lipid levels with appropriate glycemic control is standard treatment. In particular, intense low-density lipoprotein cholesterol (LDL-C) is often recommended for DM management. Based on this guidance, statins are considered the first-line lipid-lowering therapeutics for the primary and secondary prevention of circulatory disease [10,11,12]. Statins are 3-hydroxy-methylglutaryl coenzyme A (HMG-CoA) reductase inhibitors and prevent converting HMG-CoA to mevalonate by reducing cholesterol synthesis [13]. However, various statin classes have been a suspected underlying cause of new onset DM (NOD) through decreasing beta cell function and increasing insulin resistance [14,15,16]. Plausible mechanisms explain that inhibition of the limiting step of HMG-CoA suppresses glucose transporter 4, which is correlated with muscle [17], and decreasing serum adiponectin [18] leads to insulin resistance. In comparison with other statins, pitavastatin showed neutral effects on NOD, even at the highest dosage (4 mg) [19,20]. Furthermore, evidence for the “adiponectin theory” about statins has been inconsistent. 

Therefore, we investigated the effects of pitavastatin on serum adiponectin, including HMW adiponectin and insulin resistance, according to a homeostatic model for insulin resistance (HOMA-IR) and a homeostasis model of beta cell function (HOMA-β), in patients with type 2 DM and dyslipidemia.

## 2. Materials and Methods

### 2.1. Study Design and Participants

This study is an open-label randomized trial based on data from an electronic medical record (EMR) system collected between September 2018 and December 2020. From the EMR data base, a total of 114 patients were treated with pitavastatin or lifestyle modification. After a screening test, 97 subjects were allocated into either a 2 mg of pitavastatin group (PITA) or a lifestyle modification group for 24 weeks with a 1:1 ratio. Recruitment, randomization, and research visits were conducted in a single center. All patients were advised to continue their usual diet and medications. Basic assessments and blood sampling were performed at baseline, 12, and 24 weeks after patients began the study. The serum levels of total and HMW adiponectin and lipid profiles were evaluated at baseline, 12, and 24 weeks. The homeostasis model of insulin resistance (HOMA-IR) was evaluated at baseline, 12, and 24 weeks. Patients tolerated the pitavastatin treatment; those between 30 and 70 years who were diagnosed with DM and dyslipidemia and who were prescribed pitavastatin therapy were enrolled. The research nurses and clinician screened patient eligibility for inclusion. Patients with <9% glycated A1c (HbA1c) and a body mass index (BMI) from 20 to 30 kg/m^2^ were eligible to enroll. Patients with fasting triglyceride (TG) < 400 mg/dL and LDL-C > 100 mg/dL were allowed. Participants were excluded for the following criteria: (1) congestive heart failure resulting in mild shortness of breath or angina and slight limitation during ordinary activities, according to New York Heart Association (NYHA) classification [21], (2) acute syndromes of cardiovascular disease such as ischemic heart disease or stroke in the preceding three months, (3) renal dysfunction (creatinine > 1.5 mg/dL or renal replacement therapy), (4) liver dysfunction (aspartate aminotransferase (AST) or alanine aminotransferase (ALT) > 80 IU/L), (5) severe DM complications including proliferative retinopathy, and (6) pregnancy or breast feeding at the time of study. All oral antidiabetics (OAD) and insulin orders were documented and reviewed during the study. Among the OADs, patients with a previous thiazolidinedione prescription were also excluded. After exclusion, a total of 93 patients were analyzed in this study.

This study used the EMRs of patients whose treatments were terminated, and thus, there was no physical or mental risk to the patients. Informed consent was obtained before data collection. This study complied with the ethical standards of the Declaration of Helsinki and was approved by the Catholic University of Korea, Catholic Medical Center, Eunpyeong St. Mary’s Hospital Institutional Review Board (IRB approval No.PC17MESI0064, 18 January 2018).

### 2.2. Outcome Measures

Patient demographic and lifestyle variables were collected and reviewed, including age, sex, height, weight, BMI, smoking status (stratified into current smokers and non-smokers, including ex-smokers), and drinking behavior (classified as non-to-moderate drinking and risky drinking; >30 g/day of alcohol). Blood chemistries including fasting glucose, HbA1c, insulin, blood urea nitrogen (BUN), creatinine, glomerular filtration rate (GFR), sodium, potassium, AST, ALT, total cholesterol (TC), TG, high-density lipoprotein cholesterol (HDL-C), and LDL-C values were evaluated. All measurements were performed with an automated blood chemistry analyzer (Toshiba 200FR; Toshiba Ltd., Tokyo, Japan). HbA1c was measured by high-performance liquid chromatography using Diabetes Control and Complications Trial-aligned methods (Tosoh-G8; Tosoh Ltd., Tokyo, Japan). The homeostatic model assessments of insulin resistance (HOMA-IR) and of beta-cell function (HOMA-β) as insulin resistance markers were calculated as follows [22]:HOMA-IR = [fasting insulin (IU/mL) × fasting plasma glucose (mmol/L)]/22.5
HOMA-β = [20 × fasting plasma insulin (IU/mL)]/[fasting plasma glucose (mmol/L) − 3.5]

Circulating total adiponectin and HMW adiponectin concentrations were measured using a commercially available enzyme-linked immunosorbent assay (ELISA) kit. The ratio of HMW adiponectin to total adiponectin (H/T ratio) was calculated by dividing the concentration of HMW adiponectin by that of total adiponectin. Serum insulin levels were also measured using an ELISA kit (ALPCO DIAGNOSTICS Ltd., Salem, NH, USA). At the time of blood collection, all patients had fasted since the previous midnight. Blood samples were collected in tubes with EDTA and then separated by centrifugation for 15 min at 1000× *g*. Plasma was stored at −20 °C in different aliquots for each assay.

### 2.3. Primary Data Analysis

Baseline variables were presented as means and standard deviations or medians and interquartile ranges for continuous variables, based on normality, and numbers and percentages for categorical variables. Demographic variables including sex, smoking, and drinking behavior were analyzed by a complex sample analysis using Pearson’s chi-squared (χ^2^) test. The characteristics of each group were compared using independent *t*-tests for continuous variables. An adjusted linear regression analysis was performed to test for significant differences between the pitavastatin and control groups. Adiponectin and HMW adiponectin status were compared using the independent *t*-test or Wilcoxon rank-sum test for continuous variables and the chi-square or exact test for categorical variables. The primary end point was an absolute change in total or HMW adiponectin concentrations. The secondary endpoint was a change in HbA1c, HOMR-IR, and HOMA-β representing insulin sensitivity. Differences in the incidence ratios were estimated with a survival analysis, using the log-rank test to evaluate the pitavastatin group and adiponectin-related insulin sensitivity. All statistical analyses were performed using SAS^®^ software version 9.4 (SAS Institute, Cary, NC, USA). A *p*-value < 0.05 was considered statistically significant.

## 3. Results

### 3.1. Baseline Characteristics of Study Subjects

The mean age and BMI of all study subjects were 58.6 ± 9.3 years and 25.2 ± 4.0 kg/m^2^, respectively. The majority of the total population was male [52 (55.9%)]. Among 93 subjects, 44 (47.3%) were categorized into the PITA group, and 49 (52.7%) subjects were categorized as controls. Comparisons of baseline characteristics between the two groups are shown in Table 1. Age, sex distribution, BMI, waist circumference, blood pressure, lipid profiles, renal function, and insulin levels were similar between the two groups. Baseline adiponectin and HMW adiponectin levels did not differ between the PITA and control groups.

### 3.2. Effect of Pitavastatin on Lipid Profile, Total Adiponectin, HMW Adiponectin, HOMA-IR, and HOMA-β

Table 2 shows the effect of pitavastatin on lipid profiles; parameters of insulin sensitivity, including HOMA-IR and HOMA-β; and adiponectin levels. During the treatment period from baseline to 12 weeks and 24 weeks of therapy, there were no significant changes in glycemic control. In lipid profiles, the additional TC and LDL-C level reductions were significantly increased in the PITA group (TC; 199.7 ± 29.4 vs. 154.8 ± 28.7 mg/dL, *p* < 0.001, LDL-C; 124.3 ± 25.7 vs. 86.9 ± 30.2 mg/dL, *p* < 0.001). We evaluated the levels of adiponectin and HMW adiponectin as primary outcomes. The differences in adiponectin and HMW adiponectin levels at the three follow-up points were not statistically significant. To evaluate the trend of adiponectin and HMW adiponectin after exposure to pitavastatin, we performed a measured ANOVA and found that adiponectin and HMW adiponectin were elevated in the PITA group, compared with the control group, but the difference was not significant. Regarding insulin sensitivity, the PITA group had lower HOMA-IR and HOMA-β levels during the follow-up period than the control group, but not significantly (HOMA-IR, *p* = 0.5921 at 12 weeks and *p* = 0.3645 at 24 weeks; HOMA-β, *p* = 0.8915 at 12 weeks and *p* = 0.7313 at 24 weeks).

### 3.3. Association between Adiponectin and HMW Adiponectin, and HOMA-IR and HOMA-β as Insulin Sensitivity Markers

We further evaluated the correlation between adipokines and insulin sensitivity. Among 93 patients, adiponectin levels were negatively correlated with HOMA-IR (β coef-ficient −0.196, *p* = 0.072) and HOMA-β (β coefficient −0.275, *p* = 0.011) levels, and only the HOMA-β relationship was significant. There were significant negative correlations be-tween HMW adiponectin and HOMA-IR (β coefficient = −0.307, *p* = 0.004) and HOMA-β (β coefficient = −0.386, *p* = 0.003). However, the correlation was not statistically significant when comparing the two groups separately (PITA versus control).

### 3.4. Total Adiponectin Changes in the PITA Group with Baseline above 15%

A sub-analysis was performed for patients who had total percent changes of adiponectin above 15% (the highest quartile) compared with baseline adiponectin before being prescribed pitavastatin (Table 3). The PITA group demonstrated greater adiponectin increase rates than the control group OR = 2.8, 95% CI 1.009–7.774). After multivariate adjustment for age, sex, BMI, baseline adiponectin, HOMA-IR, and HOMA-β (model 2), the ORs (95% CI) were 4.86 (1.292–18.275).

## 4. Discussion

This open-label randomized trial using EMR data showed that pitavastatin was not significantly associated with adipokines changes in DM patients during 24 weeks of follow-up. However, we demonstrated that the 15% adiponectin increase compared with baseline was significantly different in the pitavastatin treatment group compared with the controls. Conversely, the results of our study indicated that pitavastatin therapy is not responsible for aggravated glucose metabolism in DM patients with dyslipidemia. There was no significant change in HbA1c, HOMA-IR, or HOMA-β. In DM patients with dyslipidemia, total adiponectin was negatively correlated with HOMA-β, and HMW levels were negatively correlated with HOMA-IR and HOMA-β.

Clinical evidence suggests that LDL-C is a crucial cause of cardiovascular disease, and that intensive LDL-C lowering therapy is strongly recommended for DM patients [10,11,23]. Previous guidelines recommend initiating a moderate-intensity statin for patients with T2DM who are aged 40–70 years but do not have established CVD. Meanwhile, a high-intensity statin is recommended for patients with T2DM and established ASCVD. However, despite these first-line recommendations of statins for dyslipidemia management, there are concerns about adverse effects of statins on NOD and dysglycemia [24,25,26]. However, the known effect of statins on dysglycemia is inconsistent according to intensity, formulation, and class [27,28], because the mechanism of statin-induced NOD is not well-established. Altered β-cell function and decreased insulin sensitivity have been suggested as possible mechanisms [29]. Among the plausible explanations for insulin sensitivity, reduced HMG-CoA reductase activity [30], suppressed glucose transporter 4 by attenuating insulin signaling [31], and decreased adipokines such as adiponectin [32] have been suggested. Therefore, we evaluated adiponectin and HMW adiponectin levels over 24 weeks. A previous study reviewed 20 clinical trials and depicted the effect of several statins including pitavastatin, simvastatin, pravastatin, fluvastatin, atorvastatin, and rosuvastatin on adiponectin [33]. Arnaboldi et al. showed that adiponectin levels increased by 27.2% after pitavastatin treatment [33]. In this study, adiponectin and HMW adiponectin levels were higher in the PITA group than the control group, whose only therapy was lifestyle modification. However, we selected DM patients with HbA1c < 9%, and baseline LDL-C was not definitely high. Therefore, there was no significant difference between the treatment and control groups. Nevertheless, after 24-weeks of follow-up, adiponectin levels were not decreased with effective lowering of LDL-C and TC. Thus, our results on the effects of pitavastatin treatment on total adiponectin concentrations in patients with T2DM and dyslipidemia are consistent with those of previous studies showing that pitavastatin does not increase adiponectin levels [33,34]. Additionally, consistent with a previous study [15], HbA1c and serum glucose levels at 12 and 24 weeks of follow-up did not differ between the PITA and control groups in this study.

The exact mechanism for the association between pitavastatin and total adiponectin or HMW adiponectin increase remains unclear. The enhancement of adipose peroxisome proliferator-activated receptor α [35] and its anti-inflammatory effects to block the production of reactive oxygen species [36] suggest that it could be a factor for adiponectin elevation.

The present study demonstrated increased HWM adiponectin levels after the 24-week pitavastatin treatment; however, there was no significance. In the total analysis, the HWM was negatively correlated with HOMR-IR or HOMA-β. Overall, we concluded that pitavastatin does not worsen glycemic status during pitavastatin treatment in DM patients.

To the best of our knowledge, this is the first study to investigate an association between pitavastatin and adipokines in Korean DM patients in a real clinical setting. Previous studies focused on pitavastatin efficacy and NOD in naïve subjects [37,38,39]. However, several limitations should be acknowledged. First, this study had a small sample size and selection bias might have been introduced from the patient exclusion process. To assuage this limitation, we strengthened the research nurse and clinician inspection at the time of enrollment. Despite a small sample size, a previous study showed significant conclusions [40]. A second weakness is the open-label design, where the investigators were aware of pitavastatin assignment. Third, a causal relationship between adiponectin and insulin sensitivity could not be confirmed. If insulin sensitivity due to pitavastatin use develops over several decades, a 24-week study is not sufficient to observe an effect. To minimize the effect of DM duration, the study excluded patients who were diagnosed with DM over 10 years prior. Larger clinical trials are needed to investigate long-term effects on glycemic control.

## 5. Conclusions

Pitavastatin treatment did not lead to increases in adiponectin or HMW adiponectin in DM patients with dyslipidemia. Although statins have been considered a risk factor for insulin resistance in DM patients, pitavastatin did not significantly decrease insulin sensitivity, according to the results of no significant change of HOMA-IR and HOMAβ. Therefore, pitavastatin could be considered in DM patients with concerns about insulin resistance and uncontrolled glycemia.

## Figures and Tables

**Table 1 jcm-11-06756-t001:** Baseline characteristics according to statin therapy.

	Pitavastatin	Control	*p*-Value
	(*n* = 44)	(*n* = 49)	
Age (year)	58.6 ± 9.6	58.6 ± 9.2	0.9652
Sex, male, *n* (%)	24 (54.5)	28 (57.1)	0.823
Weight (kg)	68.9 ± 13.1	67.4 ± 10.8	0.5264
Height (cm)	170.7 ± 43.9	162.7 ± 8.5	0.2128
BMI (kg/m^2^)	25.0 ± 4.9	25.4 ± 3.2	0.6466
Waist circumference (cm)	88.1 ± 8.1	86.6 ± 9.6	0.4261
SBP (mmHg)	121.4 ± 17.2	126.4 ± 13.2	0.1193
DBP (mmHg)	69.6 ± 9.9	77.1 ± 9.6	0.0003
HbA1c (%)	6.7 ± 0.7	6.6 ± 0.6	0.5133
Fasting glucose (mg/dL)	134.4 ± 26.6	135.0 ± 23.8	0.9079
Total cholesterol (mg/dL)	210.3 ±28.0	207.5 ± 19.6	0.5771
HDL-cholesterol (mg/dL)	48.2 ± 10.4	47.6 ± 9.2	0.7738
LDL-cholesterol (mg/dL)	137.0 ± 21.0	132.0 ± 15.8	0.2078
Creatinine (mg/dL)	0.84 ± 0.2	0.81 ± 0.2	0.578
eGFR (mL/min/1.73 m^2^)	87.7 ± 16.3	88.0 ± 16.6	0.9352
Insulin (μIU/mL)	25.1 (17.8–35.4)	23.9 (17.73–32.12)	0.8217
HOMA-β (%)	130.2 (91.1–186.1)	125.8 (91.5–172.8)	0.8843
HOMA-IR	8.2 (5.7–11.6)	7.8 (5.8–10.6)	0.8618
Triglyceride (mg/dL)	135.4 (115.1–159.3)	151.6 (130.9–175.5)	0.2994
AST (U/L)	21.6 (19.8–23.7)	23.7 (21.6–26.0)	0.1561
ALT (U/L)	22.7 (19.7–26.1)	23.5 (20.7–26.7)	0.7003
Adiponectin (μg/mL)	5.0 (4.2–6.1)	4.6 (3.8–5.6)	0.5275
HMW Adiponectin (μg/mL)	3.3 (2.57–4.16)	2.8 (2.16–3.6)	0.3414

Data are mean ± standard deviation or geometric mean (95% C.I). ALT, alanine aminotransferase; AST, aspartate aminotransferase; BMI, body mass index; DBP, diastolic blood pressure; eGFR, estimated glomerular filtration rate; HbA1c, glycated A1c; HDL-C, high-density lipoprotein cholesterol; HMW, high molecular weight; HOMA-β, homeostasis model assessment of beta cell function, HOMA-IR, homeostasis model assessment of insulin resistance; LDL-C, low-density lipoprotein cholesterol; SBP, systolic blood pressure.

**Table 2 jcm-11-06756-t002:** The comparison of lipid profile, HOMA-IR, total adiponectin, HMW adiponectin, and after pitavastatin treatment during 24 weeks of follow-up.

	Baseline	*p*-Value	12 Weeks	*p*-Value	24 Weeks	*p*-Value
HA1c (%)		0.5133		0.5256		0.5806
Control	6.6 ± 0.6		6.6 ± 0.7		6.7 ± 0.8	
Pitavastatin	6.7 ± 0.7		6.7 ± 1.2		6.8 ± 0.9	
Glucose (mg/dL)		0.9079		0.5349		0.3104
Control	135.0 ± 23.8		135.9 ± 26.3		142.0 ± 30.4	
Pitavastatin	134.4 ± 26.6		132.4 ± 25.9		135.5 ± 28.0	
TC (mg/dL)		0.5771		<0.0001		<0.0001
Control	207.5 ± 19.6		195.5 ± 30.0		199.7 ± 29.4	
Pitavastatin	210.3 ± 28.0		159.0 ± 27.9		154.8 ± 28.7	
TG (mg/dL) *		0.2994		0.2677		0.0691
Control	151.6 (130.9–175.5)		139.9 (119.4–164.0)		148.0 (127.0–172.4)	
Pitavastatin	135.4 (115.1–159.3)		123.5 (105.5–144.7)		121.6 (104.8–141.2)	
HDL-C (mg/dL)		0.7738		0.3014		0.482
Control	47.6 ± 9.2		46.3 ± 10.4		47.2 ± 10.0	
Pitavastatin	48.2 ± 10.4		48.4 ± 8.2		48.8 ± 10.7	
LDL-C (mg/dL)		0.2078		<0.0001		<0.0001
Control	132.0 ± 15.8		124.0 ± 25.7		124.3 ± 25.7	
Pitavastatin	137.0 ± 21.0		89.1 ± 26.76		86.9 ± 30.3	
HOMA-β (%)		0.8843		0.8915		0.7313
Control	125.8 (91.5–172.8)		123.4 (91.4–166.6)		114.6 (81.9–160.4)	
Pitavastatin	130.2 (91.1–186.1)		119.12 (76.7–185.1)		105.07 (71.5–154.5)	
HOMA-IR		0.8618		0.5921		0.3645
Control	7.8 (5.8–10.6)		7.7 (5.8–10.2)		8.0 (5.7–11.1)	
Pitavastatin	8.2 (5.7–11.7)		6.7 (4.5–10.1)		6.3 (4.3–9.5)	
Adiponectin (µg/mL) *		0.5275		0.4306		0.4198
Control	4.6 (3.8–5.6)		4.2 (3.5–5.1)		4.2 (3.5–5.0)	
Pitavastatin	5.0 (4.2–6.1)		4.7 (3.9–5.7)		4.7 (3.8–5.7)	
HMW		0.3414		0.417		0.2937
Control	2.8 (2.2–3.6)		2.0 (1.5–2.6)		2.0 (1.6–2.7)	
Pitavastatin	3.3 (2.6–4.2)		2.3 (1.8–3.0)		2.5 (1.9–3.2)	

* Log transformation. TG and LDL-C at baseline, 12 weeks, and 24 weeks showed significant differences. (*p*-vales < 0.001).

**Table 3 jcm-11-06756-t003:** Total adiponectin changes in PITA group with baseline above 15%.

	*n* (%)	OR (95% C.I.)		
		Crude	Model 1	Model 2
Control	7 (14.3)	1 (Ref.)	1 (Ref.)	1 (Ref.)
Pitavastatin	14 (31.8)	2.8 (1.009–7.774)	4.951 (1.411–17.378)	4.86 (1.292–18.275)

Model 1: Adjustment for age, sex, body mass index, and baseline total adiponectin. Model 2: Adjustment for age, sex, body mass index, baseline total adiponectin, and baseline HOMA-IR.

## Data Availability

All data generated or analyzed during this study are included in this published article.

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
