# Peer review of "Does Pitavastatin Therapy for Patients with Type 2 Diabetes and Dyslipidemia Affect Serum Adiponectin Levels and Insulin Sensitivity?"

_jcm, 2022, doi:10.3390/jcm11226756_

Round 1
Reviewer 1 Report
The paper submitted for the review entitled „Effect of pitavastatin therapy on serum adiponectin and insulin sensitivity in patients with type 2 diabetes and dyslipidemia” is an original and novel survey study investigating the association between pitavastatin use and adiponectin levels and insulin sensitivity in patients with type 2 diabetes and dyslipidaemia. The novelty of the study may be evidenced by the lack of publication in the available literature after the following terms were listed together:: „pitavastatin”, „type 2 diabetes mellitus”, „hiperlipidemia”, „adiponectin”, „insulin sensitivity”. The subject matter undertaken by the authors is important and useful in diabetes practice.
According to my assessment, the weaknesses of the work include:
Major issues
In the abstract, the aim of the study is incomplete, as it does not include the effect of pitavastatin on insulin sensitivity.
Conclusion in the abstract: „The present study showed a significant increased from baseline in serum adiponectin following pitavastatin therapy” is inconsistent with the results obtained.
For clarity of randomisation selection, state how many patients were included in the trial based on data from the electronic medical records system, from which 2 groups with pitavastatin and a control group were selected - after taking into account inclusion and exclusion criteria
It is evident from the text of the paper that the gender of the study patients was included in the study groups, but the number of male and female subjects was not included in Table 1 and the text
Table 2 shows that the TC and LDL-C values given in the text refer to the control group and not the PITA group (lines 154-157).
Lines 180 and 181 record „Subanalysis was performed for patients whose changes in total adiponectin were greater than 15% compared with baseline adiponectin before the prescription of pitavastatin (Table 3).” It should be explained why 15% was used as the cut-off value.
Conclusion notes: the sections of content on other statins considered to be a risk factor for NOD and the linking of insulin sensitivity to LDL-C and TG lowering should be excluded as this was not the focus of this study. It should be added that the conclusions relate to type 2 diabetes.
Minor issues
In my opinion, a more appropriate title for the work would be: „Does pitavastatin therapy for patients with type 2 diabetes and dyslipidaemia affect serum adiponectin levels and insulin sensitivity?”
In some places in the study it is recorded that the study was carried out after 12 and 24 weeks and in other places that it was carried out after 3 and 6 months. This should be standardised.
The statistical significance of the differences between the baseline values of the indicators studied and those obtained after 12 and 24 weeks of follow-up should be included in Table 2 and/or in the commentary.
Table 2 lacks an explanation of the abbreviations used and the * sign next to (ng/ml). What do Log Adiponectin and Log HMW mean?
Lines 176 and 177 record that the β-value for HOMA-IR and HOMA-β is the same. Is this an error or coincidence?
Line 198-200 recorded „In DM patients with dyslipidemia, total adiponectin and HMW levels were negatively correlated with HOMA-IR and HOMA-β.” However, a significantly negative correlation was only observed between adiponectin and HOMA-β levels.
In the discussion section is the sentence "However, we selected DM patients with HbA1c less than 10%...". (line 220). In contrast, the inclusion criteria for patients in the study state that the HbA1c value was less than 9%.
Author Response
We appreciate your efforts for reviewing our manuscript. In the revised manuscript, we made corrections and revised manuscript according to reviewers' comments and suggestions. A response to the referees΄ suggestions has been listed one by one and we attached file.

Reviewer 2 Report
Review for the manuscript entitled “Effect of pitavastatin therapy on serum adiponectin and insulin sensitivity in patients with type 2 diabetes and dyslipidemia”.
This study examined the effect of pitavastatin on adiponectin levels and insulin sensitivity in patients with type 2 diabetes. This was a negative study, and pitavastatin had no effect on adiponectin. Nevertheless, the authors conclude as foloows: The present study showed a significant increased from baseline in serum adiponectin following pitavastatin therapy.
The logical development is unclear. The statistical analysis method employed is also inappropriate. It does not meet the requirements of a scientific article.
In Table 2, changes in each parameter over time are shown for each of the two groups. The usual analysis method (main-effect/interaction P) is not used here. It is not meaningful to compare the two groups at each time point. The Pitavastatin group does not appear to show an increase in adiponectin (5.0 to 4.7 to 4.7).
Even though Table 2 shows no significant increase in adiponectin, Table 3 shows that pitavastatin contributed to the increase in adiponectin. Consistency is lacking.
Ref 38 is a review article on the effect of various statins on adiponectin. This paper shows that statins have already been shown to increase adiponectin. It seems less significant to test the same thing in a small enrollment limited to Koreans.
That’s all.
Author Response

(The authors gave the same response as above.)

Round 2
Reviewer 2 Report
The authors appear to have addressed the comments appropriately.